# Optimizing the Pore Structure of Lotus-Type Porous Copper Fabricated by Continuous Casting

**DOI:** 10.3390/ma17205015

**Published:** 2024-10-14

**Authors:** Byung-Sue Shin, Soong-Keun Hyun

**Affiliations:** Department of Materials Science and Engineering, Inha University, Incheon 22212, Republic of Korea; 22202189@inha.edu

**Keywords:** lotus-type porous copper, continuous casting, composite process parameter, pore structure, escapement coefficient

## Abstract

Lotus-type porous copper was fabricated using a continuous casting method in pressurized hydrogen and nitrogen gas atmospheres. This study evaluates the effects of process parameters, such as the hydrogen ratio, total pressure, and transference velocity, on the resulting pore structure. A continuous casting process was developed to facilitate the mass production of lotus-type porous copper. To achieve the desired porosity and pore diameter for large-scale manufacturing, a systematic evaluation of the influence of each process parameter was conducted. Lotus-type porous copper was produced within a hydrogen ratio range of 25–50%, a transference velocity range of 30–90 mm∙min^−1^, and a total pressure range of 0.2–0.4 MPa. As a result, the porosity ranged from 36% to 55% and the pore size varied from 300 to 1500 µm, demonstrating a wide range of porosities and pore sizes. Through process optimization, it is possible to control the porosity and pore size. The hydrogen ratio and total pressure were found to primarily affect porosity, whereas the hydrogen ratio, transference velocity, and total pressure significantly influenced pore diameter. When considering these parameters together, porosity was most influenced by the hydrogen ratio, whereas the total pressure and transference velocity had a greater influence on pore diameter. Reducing the hydrogen ratio and increasing the transference velocity and total pressure reduced the pore diameter and porosity. This optimization of the continuous casting process enables the control of porosity and pore diameter, facilitating the production of lotus-type porous copper with the desired pore structures.

## 1. Introduction

Lotus-type porous metals [1], also known as Gasar metals [2,3], are produced by exploiting the difference in gas solubility between the liquid and solid states of metals. When a gas dissolved in liquid metal at high temperature transitions is released upon solidification due to the solubility difference, it forms bubbles and solidifies with the metal, resulting in a regular cylindrical pore structure. Copper is commonly used to form these structures, with hydrogen gas playing a crucial role as a pore-forming agent [1,2,3]. As copper solidifies from its liquid state, the dissolved hydrogen is released, forming a characteristic porous structure. The unique properties of lotus-type porous copper, including its significantly larger specific surface area compared to conventional copper, make it a material of great interest for various industrial applications, including catalysis, filtration, and energy storage [1,4,5,6,7].

A recent study by Kim et al. provided compelling evidence of the industrial potential of lotus-type porous copper. The study confirmed the superior bond strength and shock absorption efficiency of metal–metal and metal–ceramic bonds attributed to the pore structure of lotus-type porous copper [8]. These results suggest that lotus-type porous copper is a promising bonding material in various industrial applications, particularly in semiconductor packaging and cooling applications. The effectiveness of lotus-type porous copper as a bonding material is determined by its pore structure, with large and regular pores contributing to the improved bond strength and shock absorption [8].

In addition, the unidirectionally aligned micropores of the lotus-type porous structure significantly increase the specific surface area, which, in turn, enhances the specific surface area and improves the heat exchange performance [9,10]. These structural characteristics are crucial for maintaining excellent cooling performance, even under high heat flow conditions. The high thermal conductivity of copper makes it an effective heat transfer material, and this property is further enhanced when combined with the porous lotus structure. As a result, lotus-type porous copper shows great promise for cooling electronic equipment and is crucial for developing efficient thermal management systems [11,12,13]. Consequently, manufacturing technologies that control the pore structure of lotus-type porous metals play a crucial role in these industrial applications.

Several techniques can be employed in fabricating lotus-type porous materials, including mold casting, centrifugal casting, and continuous zone melting [1,14,15,16]. Continuous casting is a suitable technique for the mass production of lotus-type porous copper [17]. Previous studies have mainly focused on evaluating lotus-type porous copper by considering only the transference velocity [17], with limited consideration of the total pressure or mixed gas conditions (N_2_ + H_2_). In addition, other studies have investigated pore structures using methods such as zone melting, mold casting, or centrifugal casting [1,15,18,19] which are based on process conditions different to continuous casting. In particular, these methods have difficulty maintaining a consistent solidification rate, making it difficult to achieve uniform porosity and pore size throughout the entire sample.

In contrast, the continuous casting process maintains a relatively stable solidification rate, enabling uniform porosity and pore size across the entire sample. However, there is a lack of systematic studies on the various process parameters related to continuous casting. Therefore, it is necessary to comprehensively evaluate the key process parameters that influence pore structure formation in continuous casting to determine which conditions have the most significant impact. This is a critical study as it provides the basis for process optimization, which is essential for efficient large-scale production.

In this study, the effects of single or composite process parameters on the pore structure of lotus-type porous copper fabricated by continuous casting were investigated. Specifically, the study examines the effects of the hydrogen ratio, which determines the amount of hydrogen introduced during the casting process, and the transference velocity, which is the speed at which molten copper is drawn through the casting apparatus. In addition, the total pressure applied during the process was analyzed to understand its effect on the porosity and pore diameter of the produced copper. The aim is to identify the optimum conditions for producing lotus-type porous copper with specific pore structures tailored to various industrial applications by systematically varying these parameters. These results provide valuable insights into the role of process control in achieving the desired pore structures.

## 2. Materials and Methods

Lotus-type porous copper was produced using a pressurized continuous casting apparatus, as shown in Figure 1. This apparatus consists of a mold connected to a crucible with a hole at the bottom, a start bar, and a dummy bar to prevent premature flow of the molten copper. The start bar had a cross-sectional area of 30 × 40 mm, corresponding to the cross-sectional area of the cast ingot. The maximum length of the solidified ingot was 1200 mm. A water-cooled copper chiller was positioned around the mold to solidify the molten copper, and a motor controlled the movement of the start bar to regulate the transference velocity.

An amount of 2 kg of pure copper (99.99 wt. % Cu) was loaded into the crucible, and the total and partial pressures of the hydrogen and nitrogen gasses were adjusted according to the experimental conditions. Both the top and bottom chambers were pressurized simultaneously to melt the copper. The crucible temperature was monitored using an R-type thermocouple, whereas the mold temperature was measured in real time using a K-type thermocouple. Once the crucible temperature reached 1523 K, it was maintained for 600 s to ensure sufficient dissolution and the diffusion of the hydrogen into the molten copper. The melting temperature, measured directly with an R-type thermocouple, was 1623 K, at which point the start bar began drawing the molten copper. The melt was cooled using a start bar connected to a mold and solidified continuously as it passed through the mold. The resulting lotus-type porous copper had a cross-sectional area of 30 × 40 mm and a maximum length of 500 mm.

The production of lotus-type porous copper involved controlling the hydrogen ratio, transference velocity, and total pressure. The detailed conditions are listed in Table 1. Although the apparatus can use 100% hydrogen gas, a mixed gas with a hydrogen gas ratio of 50% was used in this experiment. The maximum available transference velocity was 100 mm/min and the maximum total pressure was 0.4 MPa.

A discharge wire cutter (AQ550L, Sodick Co., Yokohama, Japan) was used to cut the fabricated specimens in directions perpendicular and parallel to the direction of continuous casting. Each section was mechanically polished using #400 to #2000 grit sandpaper, followed by a final polish with 1 μm diamond paste. The cross-sections of the polished specimens were then examined using a light microscope (VHX-7000, Keyence Co., Osaka, Japan). The pore diameter, density, and distribution were measured using an image analyzer (Image-Pro Plus 6.1, Media Cybernetics Co., Rockville, MD, USA). Porosity was evaluated using the following equation [1,17,18].
(1)$$\mathrm{Porosity}\;\left(\%\right)=\left(1-\frac{Apparent\;density\;of\;porous\;copper}{Density\;of\;nonporos\;copper}\right)\times 100$$

The porosity was calculated by measuring the apparent density, weight, and volume of the specimens according to the process parameters.

## 3. Results

### 3.1. Single Process Parameter

#### 3.1.1. Effect of Single Process Parameter on Porosity

Porosity was analyzed separately for different process parameters to examine their effect on the pore structure. Figure 2a show the impact of transference velocity on porosity. When the hydrogen ratio and total pressure were kept constant, the porosity remained unchanged, regardless of the transference velocity. Figure 2b show the effects of the total pressure on porosity. When the transference velocity and hydrogen ratio were kept constant, the porosity decreased as the total pressure increased. Similarly, Figure 2c show the effect of the hydrogen ratio on porosity at constant transference velocity and total pressure. In these cases, porosity generally increased with the hydrogen ratio. However, as shown in Figure 2c, at hydrogen ratios of 25% and 34%, the porosity at 0.2 MPa was lower than the theoretically predicted porosity.

Figure 3 shows the results of the pore morphology and diameter measurements under low-porosity conditions for hydrogen ratios of 25% and 34% at a total pressure of 0.2 MPa, as shown in Figure 2c. At a total pressure of 0.3 MPa, the pore diameter gradually increased with the hydrogen ratio. Conversely, at a total pressure of 0.2 MPa, the pore diameter decreased as the hydrogen gas ratio increased. In particular, the number of detected pores was significantly reduced at hydrogen ratios of 25% and 34%. In addition, the skin layer [1,17] typically observed in lotus-type porous copper was not present under these conditions.

The internal pressure of a bubble increases as it expands and decreases as it is released. As the pressure inside the bubble increases, the external pressure exerted on the melt decreases. Consequently, in low-pressure environments, pores are more likely to expand and form larger pores [20]. During this process, pores tend to merge, resulting in an overall increase in pore diameter and a decrease in porosity.

#### 3.1.2. Effect of Single Process Parameter on Pore Diameter

The effects of process parameters on pore morphology and diameter were evaluated under different process conditions. Figure 4a shows a cross-sectional pore morphology against the hydrogen ratio, with transference velocity and total pressure kept constant. The pore diameter tended to increase with hydrogen ratio. Figure 4b shows the pore diameter measurements, consistent with the trend observed in Figure 4a. The pore morphology and diameter relative to the transference velocity are shown in Figure 4c,d, respectively. As transference velocity increased, the pore diameter decreased. The pore morphologies and diameters corresponding to the total pressure are shown in Figure 4e,f, respectively. An increase in total pressure resulted in a decrease in pore diameter.

### 3.2. Composite Process Parameter

The effects of two or three process parameters on pore structure were analyzed by varying these parameters in different experimental sets. In each experiment, either one of the parameters (hydrogen fraction, transference velocity, or total pressure) was fixed while the other parameter was varied, or all three parameters were varied simultaneously.

#### 3.2.1. Effect of Composite Process Parameter on Porosity

Porosity is influenced by both the hydrogen ratio and total pressure, as shown in Figure 2. As the hydrogen ratio increased, porosity also increased, whereas an increase in total pressure decreased porosity. When both the hydrogen ratio and total pressure were simultaneously increased, porosity increased, as shown in Figure 5.

This suggests that the effect of increasing porosity due to the hydrogen ratio was more dominant than the effect of decreasing porosity due to the total pressure. Therefore, careful consideration of the interaction between the hydrogen ratio and total pressure is necessary for optimizing porosity. In particular, since the porosity-increasing effect of a higher hydrogen ratio exceeds the porosity-decreasing effect of an increased total pressure, achieving the optimal porosity requires a balanced adjustment of these parameters. Consequently, proper control of the hydrogen ratio is a critical factor in determining porosity.

#### 3.2.2. Effect of Composite Process Parameter on Pore Diameter

Increasing the hydrogen ratio leads to an increase in pore diameter. Conversely, increasing the transference velocity decreases the pore diameter while increasing the number of detected pores. Consequently, while the porosity is influenced exclusively by the hydrogen ratio, pore diameter is influenced by two opposing process parameters: the hydrogen ratio and transference velocity. Figure 6 shows the variation in pore diameter under combined process conditions. Figure 6a shows the cross-section of the pores as the hydrogen ratio and transference velocity increase, with the total pressure kept constant. As both the hydrogen ratio and transference velocity increase, the pores decrease in diameter and increase in number. The measurement results are shown in Figure 6b. The tendency for pores to decrease in diameter and increase in number with increasing transference velocity is more significant than the tendency for pores to increase in diameter with an increasing hydrogen ratio.

The results of increasing the hydrogen ratio and total pressure while maintaining a constant transference velocity are shown in Figure 7. At 0.2 MPa, the porosity and pore diameter were not considered for the low hydrogen ratio, as the low-pressure process shown in Figure 3 caused coarsening due to pore expansion. The results in Figure 7a show that pore diameter decreases, and the number of detected pores increases, as shown in Figure 7b. The reduction in pore diameter due to increasing total pressure is more significant than the increase associated with a higher hydrogen ratio.

The results of increasing the transference velocity and total pressure while maintaining a constant hydrogen ratio are shown in Figure 8. As the transference velocity and total pressure increase, the pore diameter decreases, and the number of detected pores increases. Comparing Figure 6 and Figure 7, it can be observed that the pore diameter decreases rapidly, and the number of detected pores increases significantly. This indicates that when the two process parameters are increased simultaneously, the decrease in pore diameter and the increase in pore number becomes more significant.

The results of decreasing the hydrogen ratio while increasing the transference velocity and total pressure to decrease the pore diameter and increase the number of detected pores are shown in Figure 9. The measured results indicate a decrease in pore diameter and an increase in the number of detected pores. As the hydrogen ratio decreases and the transference velocity and total pressure increase, porosity also decreases. This is because the hydrogen ratio has the most significant effect on porosity, and a reduction in porosity also leads to a decrease in pore diameter. The combination of these three process parameters to reduce the pore diameter further accentuated the trend of decreasing pore diameter and increasing pore number. Therefore, reducing the hydrogen ratio while increasing transference velocity and total pressure effectively reduces pore diameter and increases the number of pores detected. However, it is essential to note that a decrease in hydrogen ratio is also associated with a reduction in porosity.

## 4. Discussion

### 4.1. Porosity

When the total pressure and hydrogen ratio in the liquid phase are sufficient, it is theoretically possible to calculate porosity. The principle of pore formation is based on the difference in hydrogen solubility between the liquid and solid phases, with porosity determined by the solubility difference [1,2,21]. However, not all hydrogen excluded from the solid is trapped in the pores, some escapes [17,21,22,23]. This escaped hydrogen is referred to as the escape coefficient [22,23] and must be considered when calculating porosity.

Park et al. [17] considered the escape coefficient and used *m* as a fitting parameter in their study. Although their production method was similar to continuous casting, they did not directly calculate the escape concentration but expressed it as an increase in solid volume. In contrast, Liu et al. [22,23] directly calculated the escape coefficient *a*, but their calculations were inconsistent due to differences in the casting method as they used mold casting. The escape concentration may vary depending on the casting method [17].

Therefore, to accurately calculate the porosity of lotus-type porous copper produced using the continuous casting method, it is necessary to calculate the escape coefficient directly. The escape coefficient is defined as the ratio of the escaped hydrogen concentration to the hydrogen concentration in the liquid phase [22,23], and is calculated as follows
(2)$$\alpha =\frac{C_{E}}{C_{L}}$$

To calculate the escape concentration of hydrogen, the concentration is configured as follows
(3)$$C_{L}=C_{H}+C_{S}+C_{E}$$
where $C_{L}$ is the hydrogen concentration in the liquid phase and $C_{H}$ is the hydrogen concentration of liquid copper at the solid–liquid interface during solidification at a supersaturated concentration. $C_{S}$ is the hydrogen concentration in solid copper, and $C_{E}$ is the escaped hydrogen concentration. The $C_{L}$ and $C_{S}$ were calculated using Equations (4) and (5), respectively, as proposed by Fromm and Gebhart [24].
(4)$$logC_{L}=0.5\cdot logP_{H}-2.35-\frac{2270}{T_{L}}\cdot (1084\sim 1550\;{}^{\circ}C)$$
(5)$$logC_{S}=0.5\cdot logP_{H}-2.65-\frac{2560}{T_{S}}\cdot (500\sim 1083\;{}^{\circ}C)$$

Generally, when a metal solidifies in a liquid phase containing a gas, the gas is rejected due to the difference in gas solubility, leading to supersaturation of the gas at the solid–liquid interface. To calculate the supersaturation concentration, the concentration in the liquid phase at the melting point ($T_{m}$) was calculated as follows
(6)$$C_{0}\left(T_{m}\right)=K_{l}\sqrt{P_{H}}$$

$K_{l}$ is the equilibrium constant according to Sieverts’ law and can be calculated using Equation (4). $P_{H}$ is the hydrogen pressure and reflects the supersaturation concentration, which can be expressed as follows [25]
(7)$$C_{H}\left(T_{m}\right)=F_{l}K_{l}\sqrt{P_{H}}$$
where $F_{l}$ is a constant representing the tendency of the melt to become supersaturated with gas. Supersaturation is established when $F_{l}\geq 1$.

Figure 10a shows the supersaturation constant ($F_{l}$) with hydrogen pressure, calculated using Equation (7). The supersaturation constant ($F_{l}$) remained constant regardless of hydrogen pressure. Figure 10b shows the difference between the liquid phase concentration, supersaturation concentration, and the sum of the solid phase concentrations relative to hydrogen pressure. The difference between these two plots represents the escape concentration, which increases with hydrogen pressure.

However, as shown in Figure 11, the escape coefficient did not change with increasing hydrogen and nitrogen pressures. This is because the escape coefficient *a* depends on the liquid temperature, not on the hydrogen pressure [21]. The pore volume is calculated using the escape coefficient to calculate the porosity [17,26].
(8)$$V_{\rho }=\frac{\left(k_{l}\cdot \left(1-\alpha \right)-k_{s}\right)\sqrt{P_{H}}\cdot R\cdot T_{m}}{P_{N}+P_{H}}$$
where $k_{s}$ is the equilibrium constant used to calculate the hydrogen concentration in the solid phase, as determined by Equation (5). R is the gas constant and $P_{N}$ is the nitrogen pressure when using a mixed gas. Using this value, the porosity was calculated as follows
(9)$$\varepsilon \approx \frac{V_{\rho }}{V_{\rho }+V_{S}}\cdot 100$$
(10)$$\varepsilon \approx \frac{\left(k_{l}\cdot \left(1-\alpha \right)-k_{s}\right)\cdot R\cdot T_{m}}{\left(k_{l}\cdot \left(1-\alpha \right)-k_{s}\right)\cdot R\cdot T_{m}+V_{S}\cdot \frac{P_{N}+P_{H}}{\sqrt{P_{H}}}}\;\cdot 100$$

The calculated porosity, shown as a dash–dot line in Figure 2, agrees with the experimental results. Although hydrogen pressure significantly affects porosity, the effect of total pressure cannot be disregarded. Therefore, both hydrogen and total pressure must be considered when setting the most effective process conditions for porosity control.

### 4.2. Pore Diameter

To understand the number and diameter of the pores, it is necessary to consider pore nucleation and growth. Pore nucleation has been extensively studied [1,6,27,28,29] and it has been reported that the Gibbs free energy for pore formation is reduced, resulting in heterogeneous pore nucleation [1,21,25,27]. The relationship between the nucleation rate *J* and the Gibbs free energy of a pore is given by [17,30]
(11)$$J=\frac{NkT}{h}exp\left(-\frac{1}{kT}\cdot \frac{16\pi }{3}\cdot \frac{\gamma^{3}}{\Delta P^{2}}\cdot f\left(\theta \right)\right)$$
where *N* is the number of gas molecules in the liquid phase, *k* is the Boltzmann constant, *T* is the temperature, *h* is Planck’s constant, *γ* is the surface energy of the pores, Δ*P* is the difference between the ambient pressure and the internal pressure of the pores, and (*θ*) is a function of the surface energy which depends on the contact angle *θ* between the solid and the pores. As the pore nucleation rate increases, the number of detected pores also increases.

Pore growth is closely related to diffusion. A supersaturated concentration forms at the solid–liquid interface due to the concentration difference caused by the different solubilities between the liquid and solid phases. This concentration difference leads to diffusion between the liquid and solid phases, driving pore growth [1,21]. The amount of gas entering the pores through the melt interface during a given time interval $t_{1}$ to $t_{2}$ is expressed as $q_{g}$ [25].
(12)$$q_{g}=\int_{t_{1}}^{t_{2}}\int D\cdot \nabla C\left(s,\tau \right)\cdot ds\cdot d\tau$$
where, *D* is the diffusion coefficient and $\nabla C\left(s,\tau \right)$ is the concentration gradient. Gas enters the pores by diffusion and, as the gas continues to diffuse, the diameter of the pores increases. The total pore volume is calculated using Equation (8), and given the diameter and number of detected pores, it is expressed as follows
(13)$$V_{\rho }=N_{\rho }\times V_{avg.}$$
where, $N_{\rho }$ is the number of detected pores, which refers to the number of pores measured on the cross-section, and $V_{avg.}$ is the average pore volume. The porosity and pore diameter of samples manufactured by continuous casting show more consistent trends compared to other manufacturing methods. As a result, the average pore volume can be estimated from the porosity and the number of detected pores. Although the elongated shape of the pores may cause slight variations in the number of pores depending on the cross-sectional position of the specimen, variations in pore size and the number of pores are minimal. If the nucleation rate of the pores dominates while the total pore volume remains constant, the number of detected pores increases and the average pore volume decreases. Conversely, if pore growth is dominant, the number of detected pores decreases and the average pore volume increases. An increase in average pore volume results in an increase in pore diameter.

According to the hydrogen ratio, the nucleation rate is related to the number of gas molecules (*N*) in the liquid phase. As the hydrogen concentration increases, the number of hydrogen molecules also increases. Therefore, the hydrogen ratio is proportional to *N*.

Pores form at the interface between the solid and liquid phases [25], with the diffusion direction aligned towards the supersaturated concentration in the liquid phase. Figure 12a shows the difference between the hydrogen concentration in the liquid phase and the supersaturated concentration. The concentration gradient tends to increase with increasing hydrogen pressure. This indicates that, at a constant transference velocity and total pressure, the concentration gradient increases with the hydrogen ratio, resulting in a more significant amount of gas entering the pores and leading to pore growth.

Figure 12b shows that the number of detected pores decreased as the hydrogen ratio increased. Although both nucleation and the growth of pores increased with increasing hydrogen ratio, the experimental results showed a decrease in the number of detected pores. This suggests that the nucleation rate dominates pore growth. Transference velocity is related to the solidification rate, and the supercooling Δ*T* during solidification is related to Δ*P* by the Clausius–Clapeyron equation. The relationship is expressed as follows [17,31]
(14)ΔP ∝ ΔT

Figure 13 shows the number of detected pores increased with the solidification rate, indicating that pore nucleation was dominant.

The relationship between the total pressure and pore nucleation is described in Equation (11), where Δ*P* is the pressure difference between the inside and outside of the pore. As the total pressure increases, Δ*P* also increases, resulting in a higher nucleation rate. Pore nucleation occurs at the solid–liquid interface in the supersaturated concentration ($C_{H}$). The internal pressure of the generated pores can be calculated using the following equation [25]
(15)$$C_{H}\left(T_{m}\right)=F_{l}\cdot K_{l}\sqrt{P_{H}}=K_{l}\cdot \sqrt{{F_{l}}^{2}\cdot P_{H}}$$
(16)$$P_{inter}={F_{l}}^{2}\cdot P_{H}$$
where $P_{inter}$ is the effective pressure, which is the internal pressure calculated at the time of pore nucleation. The total pressure, an experimental process parameter, serves as the external pressure during pore formation. The differences between internal and external pore pressures at a constant hydrogen ratio and transference velocity are shown in Figure 14a. As the total pressure increases, the difference between the internal and external pressures also increases. Consequently, the nucleation rate of the pores increases with increasing total pressure. The relationship between total pressure and pore growth is related to the diffusion coefficient (*D*) and can be summarized as follows [32]
(17)$$D=yn\prime v\exp\left(-\frac{cBQ}{kT}\right)$$
where *y* is the correlation factor, *n*′ is the lattice constant, *v* is the frequency factor, *c* is a constant, *B* is the bulk modulus, *Q* is the average volume per atom, *k* is the Boltzmann constant, and *T* is the temperature. The bulk modulus represents the resistance of a material to compression by external pressure. As the total pressure increases, the bulk modulus also increases, decreasing the diffusion coefficient and reducing the amount of gas entering the pores, thus inhibiting pore growth. Figure 14b shows that the number of pores detected increases as the total pressure increases. This increase in total pressure suggests that pore nucleation is dominant.

The effects of the process parameters on pore formation were confirmed, demonstrating that an increase in the hydrogen gas ratio promotes both pore nucleation and growth, with pore growth being more dominant. Conversely, an increased transference velocity and total pressure favor nucleation over pore growth.

## 5. Conclusions

In this study, the effects of single and composite process parameters, hydrogen ratio, transference velocity, and total pressure, on the pore structure of lotus-type porous copper fabricated by continuous casting were systematically investigated. The results are summarized as follows:Porosity control: Hydrogen ratio was identified as a critical factor influencing porosity. Higher hydrogen ratios resulted in increased porosity as more hydrogen gas contributed to pore formation. Conversely, increasing the total pressure decreased porosity by inhibiting pore expansion. When both hydrogen ratio and total pressure were increased simultaneously, the porosity-increasing effect of the hydrogen ratio was more dominant than the porosity-decreasing effect of the total pressure. Therefore, achieving the desired porosity requires a careful balance between these two parameters.Pore diameter and number: Pore diameter increased at higher hydrogen ratios due to increased pore growth. However, increasing the transference velocity and total pressure caused the pore diameter to decrease while the number of detected pores increased. This indicates that higher transference velocities and total pressures promote pore nucleation over pore growth. The internal pressure of the pores increases with hydrogen concentration, while the external pressure applied during the process reduces pore expansion, resulting in smaller pores and higher pore density.Pore nucleation and growth: Pore nucleation and growth are affected by several factors. Increasing the hydrogen ratio enhances both pore nucleation and growth, but the experimental results suggest that pore growth is more dominant under these conditions. On the other hand, increasing the transference velocity and total pressure favored nucleation over growth, resulting in a greater number of smaller pores. This balance between pore nucleation and growth is critical for controlling pore size and distribution.Process parameter optimization: The optimization of process parameters is essential to controlling the pore structure of lotus-type porous copper. To achieve the smallest pore size, a hydrogen ratio of 25%, a feed rate of 90 mm/min, and a total pressure of 0.4 MPa resulted in a porosity of about 36% and a pore size of 300 µm. By systematically adjusting the process parameters within the ranges of a hydrogen ratio of 25–50%, a transference velocity of 30–90 mm/min, and a total pressure of 0.2–0.4 MPa, it is possible to precisely control the porosity (36–55%) and pore size (300–1500 µm) of lotus-type porous copper. This makes it possible to produce materials with properties tailored to specific requirements.

## Figures and Tables

**Figure 1 materials-17-05015-f001:**
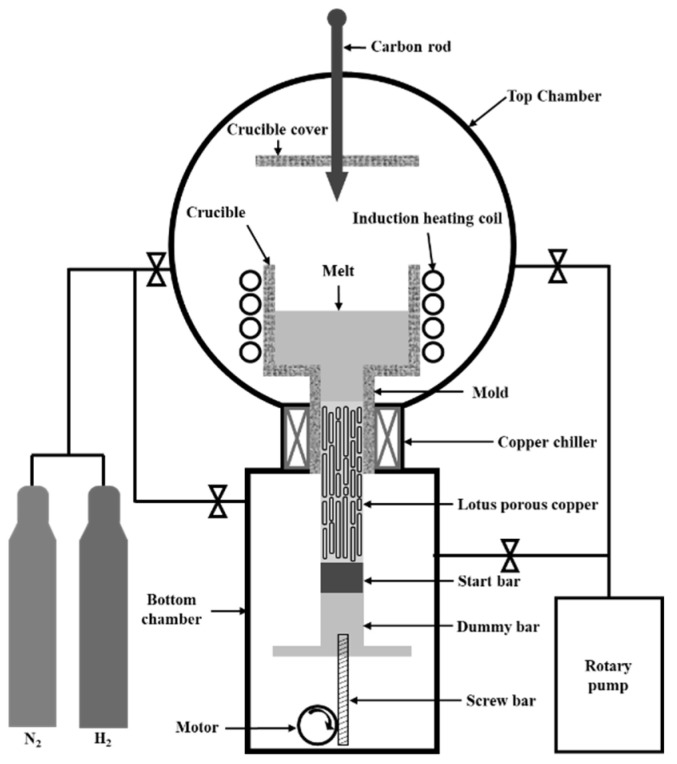
Schematic of the continuous casting apparatus.

**Figure 2 materials-17-05015-f002:**
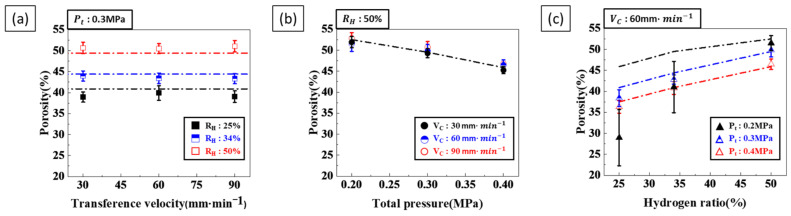
Porosity as a function of process parameters (dash–dot lines represent porosity calculated by Equation (10)): (**a**) Porosity versus transference velocity at a total pressure of 0.3 MPa, with hydrogen ratios of 25%, 34%, and 50%, respectively. (**b**) Porosity versus total pressure at a hydrogen ratio of 50%, with transference velocities of 30, 60 and 90 $\mathrm{mm}\cdot \min^{-1}$, respectively. (**c**) Porosity versus hydrogen ratio at a transference velocity of 60 $\mathrm{mm}\cdot \min^{-1}$, with total pressures of 0.2 MPa, 0.3 MPa, and 0.4 MPa, respectively.

**Figure 3 materials-17-05015-f003:**
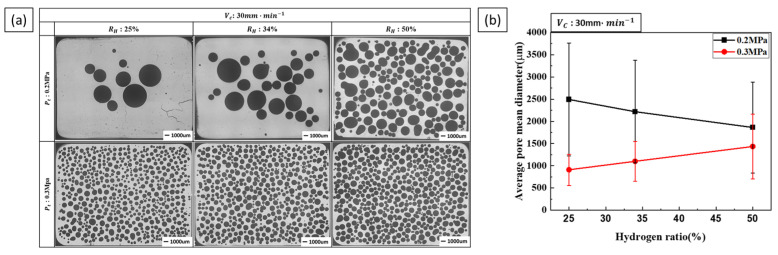
Pore morphology and size as a function of the total pressure and hydrogen ratio at a transference velocity of 30 $\mathrm{mm}\cdot \min^{-1}$: (**a**) the perpendicular cross-section and (**b**) the measured pore diameter.

**Figure 4 materials-17-05015-f004:**
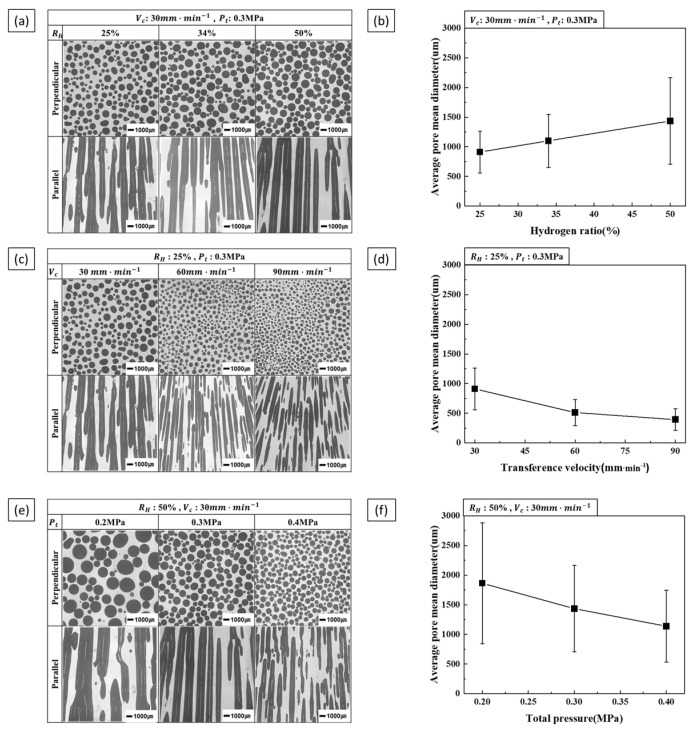
Pore morphology and size as a function of process parameters: (**a**) the perpendicular and parallel cross-sections, (**b**) the measured pore diameter as a function of hydrogen ratio at a transference velocity of 30 $\mathrm{mm}\cdot \min^{-1}$ and a total pressure of 0.3 MPa, (**c**) the perpendicular and parallel cross-sections, (**d**) the measured pore diameter as a function of transference velocity at a hydrogen ratio of 25% and a total pressure of 0.3 MPa, (**e**) perpendicular and parallel cross-sections, (**f**) the measured pore diameter as a function of total pressure at a hydrogen ratio of 50% and a transference velocity of 30 $\mathrm{mm}\cdot \min^{-1}$.

**Figure 5 materials-17-05015-f005:**
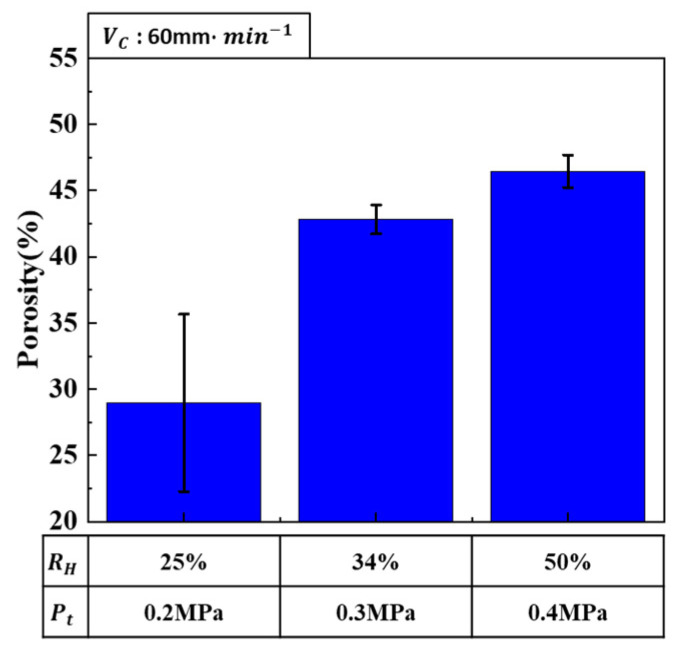
Porosity as a function of combined hydrogen ratio and total pressure at a transference velocity of 60 $\mathrm{mm}\cdot \min^{-1}$.

**Figure 6 materials-17-05015-f006:**
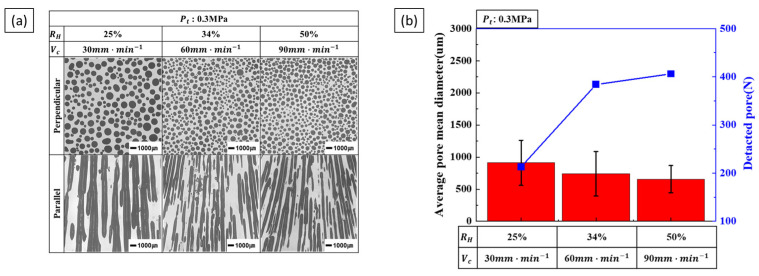
Pore morphology and size as a function of the combined hydrogen ratio and transference velocity at a total pressure of 0.3 MPa: (**a**) the perpendicular and parallel cross-sections and (**b**) the detected pores and measured pore diameters.

**Figure 7 materials-17-05015-f007:**
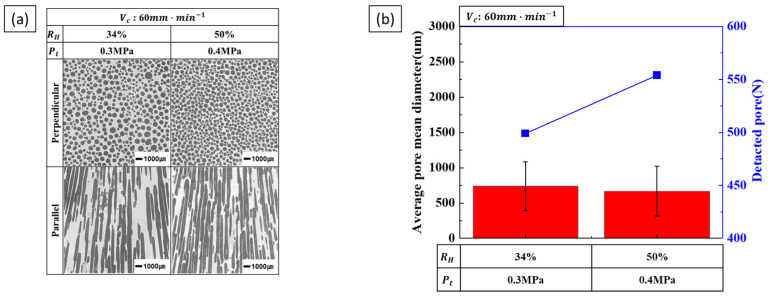
Pore morphology and size as a function of the combined hydrogen ratio and total pressure at a transference velocity of 60$\;\mathrm{mm}\cdot \min^{-1}$: (**a**) the perpendicular and parallel cross-sections and (**b**) the detected pores and measured pore diameter.

**Figure 8 materials-17-05015-f008:**
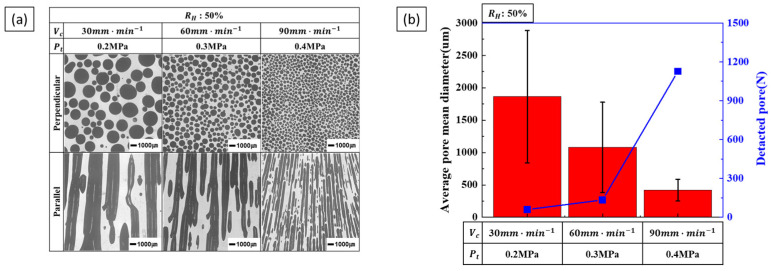
Pore morphology and size as a function of combined transference velocity and total pressure at a hydrogen ratio of 50%: (**a**) the perpendicular and parallel cross-sections and (**b**) the detected pores and measured pore diameter.

**Figure 9 materials-17-05015-f009:**
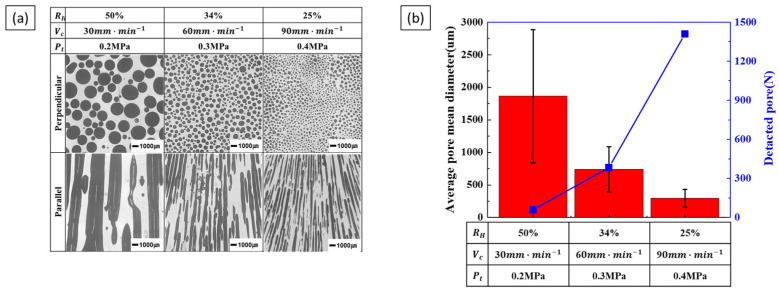
Pore morphology and size as a function of combined process parameters: (**a**) the perpendicular and parallel cross-sections and (**b**) the detected pores and measured pore diameter.

**Figure 10 materials-17-05015-f010:**
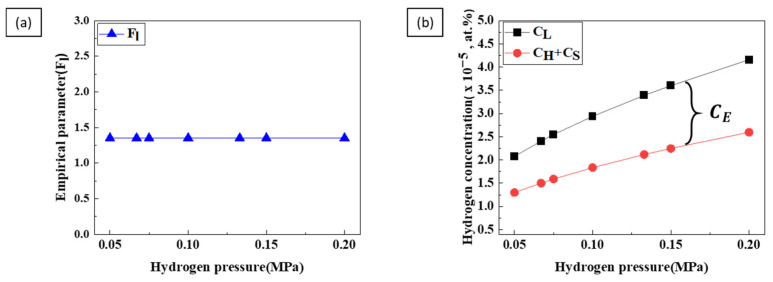
(**a**) The supersaturation constant as a function of hydrogen pressure and (**b**) the concentration in the liquid phase and total supersaturation concentration (including the solid phase) as a function of hydrogen pressure.

**Figure 11 materials-17-05015-f011:**
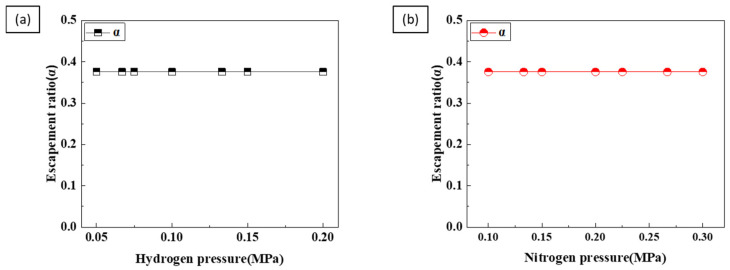
The escape coefficient as a function of (**a**) hydrogen pressure and (**b**) nitrogen pressure.

**Figure 12 materials-17-05015-f012:**
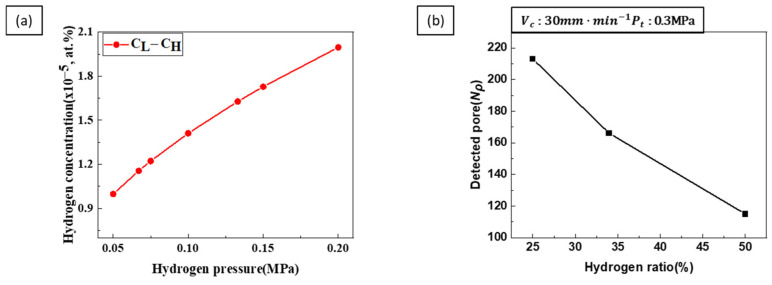
(**a**) The difference between the liquid phase and supersaturated hydrogen concentrations as a function of hydrogen pressure. (**b**) The number of detected pores was measured as a function of hydrogen ratio at a transference velocity of 30$\;\mathrm{mm}\cdot \min^{-1}$ and a total pressure of 0.3 MPa.

**Figure 13 materials-17-05015-f013:**
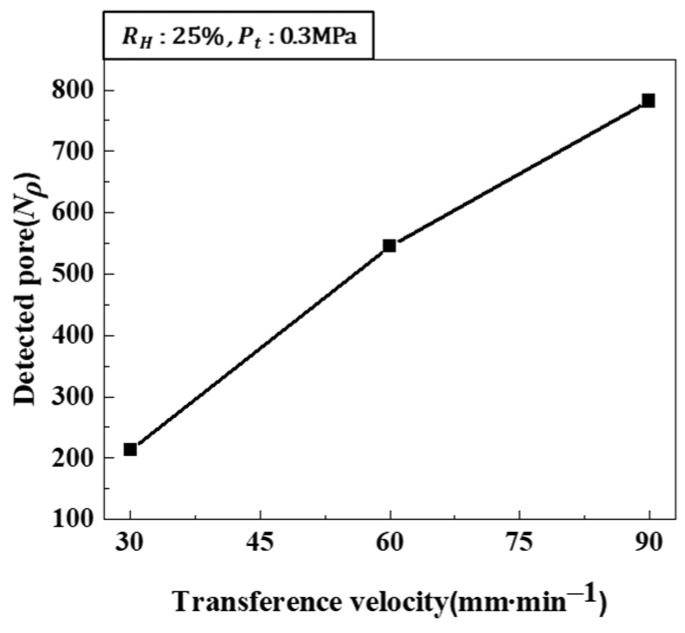
The number of detected pores was measured as a function of transference velocity at a hydrogen ratio of 25% and a total pressure of 0.3 MPa.

**Figure 14 materials-17-05015-f014:**
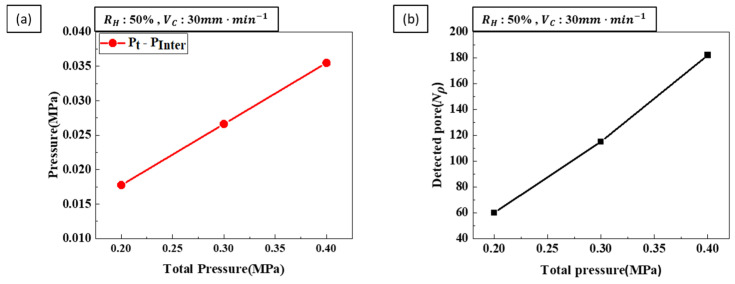
(**a**) The pressure difference between the internal and external pore environments and (**b**) the number of detected pores was measured as a function of total pressure at a hydrogen ratio of 50% and a transference velocity of 30$\;\mathrm{mm}\cdot \min^{-1}$.

**Table 1 materials-17-05015-t001:** Process parameters used in the experiment.

No.	Total Pressure (MPa)	Hydrogen Ratio (%)	Transference Velocity (mm·min^−1^)	No.	Total Pressure (MPa)	Hydrogen Ratio (%)	Transference Velocity (mm·min^−1^)	No.	Total Pressure (MPa)	Hydrogen Ratio (%)	Transference Velocity (mm·min^−1^)
1	0.2	25	30	10.	0.3	25	30	19.	0.4	25	30
2			60	11.			60	20.			60
3			90	12.			90	21.			90
4		34	30	13.		34	30	22.		34	30
5			60	14.			60	23.			60
6			90	15.			90	24.			90
7		50	30	16.		50	30	25.		50	30
8			60	17.			60	26.			60
9			90	18.			90	27.			90

## Data Availability

The raw data supporting the conclusions of this article will be made available by the authors on request.

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
