# Peer review of "Optimizing the Pore Structure of Lotus-Type Porous Copper Fabricated by Continuous Casting"

_materials, 2024, doi:10.3390/ma17205015_

Round 1
Reviewer 1 Report
Comments and Suggestions for Authors
This paper analyzed the effects of various process parameters on the pore structure of lotus porous copper produced by continuous casting. The individual parameters on the pore structure were presented with analysis in the discussion part. However, there are still following points to be clairified:
1. The background or novelity of present work is not well presented in the introduction. As mentioned in L56-L64, there are already numbers of researches working on these parameters during the continuous casting. Then what's the research necessarity of present work?
2. It is recommendated to join Fig. 2 a-c into one for better comparation. Same to d-f and g-i.
3. L133-134: "However, as shown in Figure 2(g), at hydrogen ratios of 25% and 34%, the porosity at 0.2 MPa was lower than the theoretically predicted porosity". How is the theorectically predicted value measured? and what's the reason for this lower value?
4. In Fig. 3b, it seems there is no different on the pore size under 0.3MPa.
5. For the conclusion, please highlight the findings into simple points. The last one is hard to be one conclusion.
Comments on the Quality of English Language
N/A
Author Response
Dear Reviewer
The first feedback taught me a lot. I really appreciate it. We considered them carefully and tried to reflect them in the revised manuscript accordingly.
Following are the summaries of our revision. I don't think all of the reviewer's opinions have been reflected. If it has not been reflected or if additional corrections are required, please kindly let us know. Looking forward to hearing from you soon.
Best regards,
Byungsue shin
E-mail: 22202189@inha.edu
Response to Reviewer 1 Comments
This paper analyzed the effects of various process parameters on the pore structure of lotus porous copper produced by continuous casting. The individual parameters on the pore structure were presented with analysis in the discussion part. However, there are still following points to be clairified:
- The background or novelity of present work is not well presented in the introduction. As mentioned in L56-L64, there are already numbers of researches working on these parameters during the continuous casting. Then what's the research necessarity of present work?
Response 1
This study focuses on the consistent mass production of lotus porous copper's pore structure using the continuous casting process.
Continuous casting is a suitable technique for the mass production of lotus porous copper [17]. Previous studies have mainly focused on evaluating lotus porous copper by considering only the transference velocity [17], with limited consideration of the total pressure or mixed gas conditions (N2 + H2). In addition, other research has investigated pore structures using methods such as zone melting or mold casting, centrifugal casting[1,15,18,19] which are based on process conditions different from continuous casting. In particular, these methods have difficulty keeping a consistent solidification rate, making it difficult to achieve uniform porosity and pore size entire sample.
In contrast, the continuous casting process keeps a relatively stable solidification rate, allowing for uniform porosity and pore size entire sample. However, there is a lack of systematic studies on various process parameters related to continuous casting. Therefore, it is necessary to comprehensively evaluate the key process parameters that influence pore structure formation in continuous casting to determine which conditions have the most significant impact. This is a critical study as it provides the basis for process optimization, which is essential for efficient large-scale production.
- It is recommendated to join Fig. 2 a-c into one for better comparation. Same to d-f and g-i.
Response 2
I have made the revisions.
- L133-134: "However, as shown in Figure 2(g), at hydrogen ratios of 25% and 34%, the porosity at 0.2 MPa was lower than the theoretically predicted porosity". How is the theorectically predicted value measured? and what's the reason for this lower value?
Response 3
In this study, the theoretically predicted porosity was calculated using Equation 11, and the corresponding values are indicated with dash-dots in Figure 2. This has been specified in lines 137 and 138.
However, at 0.2 MPa with hydrogen ratios of 25% and 34%, the experimentally observed porosity was lower than the theoretically calculated values. This discrepancy is likely due to the interaction between total pressure and hydrogen partial pressure during the experimental process. Specifically, when the total pressure is high, hydrogen diffuses sufficiently into the molten metal even with low hydrogen partial pressure. In contrast, when the total pressure is low, insufficient hydrogen diffusion occurs in the molten metal at low hydrogen partial pressure, leading to lower porosity than theoretically predicted. No references or studies under similar conditions were found, and the results of this study are based on experimental observations.
Therefore, to verify this phenomenon, more precise experimental designs are required under conditions where both total pressure and hydrogen partial pressure are low. We are currently designing new experiments to verify this observation and are preparing an additional paper with the relevant findings.
As such, this information has not been included.
- In Fig. 3b, it seems there is no different on the pore size under 0.3MPa.
Response 4
In this study, we analyzed the changes in pore structure that occur with increasing hydrogen ratio. In general, the number of pores decreases as the hydrogen ratio increases, while the porosity and pore size tend to increase. This trend was confirmed by the observation that both pore size and cross-sectional porosity increased with increasing hydrogen ratio under 0.3 MPa pressure.
However, opposite results were observed under the 0.2 MPa condition. As the hydrogen ratio increased, the number of pores and the porosity increased, but the pore size actually decreased. This phenomenon is related to the coalescence of pores at lower hydrogen partial pressures. When the hydrogen partial pressure is sufficiently high, hydrogen diffuses well into the molten metal, resulting in more evenly distributed pore formation across the cross section. In contrast, when the hydrogen content was 25% or 34% at 0.2 MPa, the hydrogen diffusion was insufficient, causing the pores to concentrate in the center rather than spreading out. As a result, the characteristic skin layer on the outer region of the lotus porous metal did not form.
Therefore, when producing lotus porous copper by continuous casting, it is suggested that a hydrogen ratio of at least 50% under a pressure of 0.2 MPa is required for the formation of a proper pore structure. On the other hand, normal lotus porous copper can be produced at a pressure of 0.3 MPa. To further substantiate the pore formation or coalescence phenomena observed under low pressure conditions of 0.2 MPa, we are currently designing new experiments and preparing an additional paper to include these results.
- For the conclusion, please highlight the findings into simple points. The last one is hard to be one conclusion.
Response 5
I have revised the conclusion overall.
In this study, the effects of single and composite process parameters, hydrogen ratio, transference velocity, and total pressure on the pore structure of lotus porous copper fabricated by continuous casting were systematically investigated. The results are summarized as follows:
- Porosity control: Hydrogen ratio was identified as a critical factor influencing porosity. Higher hydrogen ratios resulted in increased porosity as more hydrogen gas contributed to pore formation. Conversely, increasing the total pressure decreased porosity by inhibiting pore expansion. When both hydrogen ratio and total pressure were in-creased simultaneously, the porosity increasing effect of the hydrogen ratio was more dominant than the porosity decreasing effect of the total pressure. Therefore, achieving the desired porosity requires a careful balance between these two parameters.
- Pore diameter and number: Pore diameter increased at higher hydrogen ratios due to increased pore growth. However, increasing the transference velocity and total pres-sure caused the pore diameter to decrease while the number of detected pores in-creased. This indicates that higher transference velocities and total pressures promote pore nucleation over pore growth. The internal pressure of the pores increases with hydrogen concentration, while the external pressure applied during the process re-duces pore expansion, resulting in smaller pores and higher pore density.
- Pore nucleation and growth : Pore nucleation and growth are affected by several factors. Increasing the hydrogen ratio enhances both pore nucleation and growth, but experimental results suggest that pore growth is more dominant under these conditions. On the other hand, increasing the transference velocity and total pressure favored nucleation over growth, resulting in a greater number of smaller pores. This balance between pore nucleation and growth is critical for controlling pore size and distribution.
- Process Parameter Optimization: Optimization of process parameters is essential to control the pore structure of lotus porous copper. To achieve the smallest pore size, a hydrogen ratio of 25%, a feed rate of 90 mm/min and a total pressure of 0.4 MPa resulted in a porosity of about 36% and a pore size of 300 µm. By systematically adjusting the process parameters within the ranges of a hydrogen ratio of 25-50%, a transference velocity of 30-90 mm/min, and a total pressure of 0.2-0.4 MPa, it is possible to precisely control the porosity (36-55%) and pore size (300-1500 µm) of lotus porous copper. This makes it possible to produce materials with properties tailored to specific requirements.

Reviewer 2 Report
Comments and Suggestions for Authors
I read the article with interest. I congratulate the authors on their research and preparation of the text.
I recommend the article for publication.
In order for other readers to better know and understand the content, meaning and results, I am attaching some comments that can be supplemented or clarified more thoroughly before publication.

Author Response
Dear Reviewer
The first feedback taught me a lot. I really appreciate it. We considered them carefully and tried to reflect them in the revised manuscript accordingly.
Following are the summaries of our revision. I don't think all of the reviewer's opinions have been reflected. If it has not been reflected or if additional corrections are required, please kindly let us know. Looking forward to hearing from you soon.
Best regards,
Byungsue shin
E-mail: 22202189@inha.edu
Response to Reviewer 2 Comments
Places in the article to which I draw attention:
Line 13-19
The statements made seem obvious in my opinion. Low hydrogen content and rapid crystallization lead to lower porosity.
I would have preferred the authors to state specifically which factor and what value results in: highest / lowest porosity, highest / lowest pore diameter.
Important information was included in the introduction. It would be worth considering adding at least one or more examples of the use of the material covered in the article.
Response 1
I have made the revisions.
Lotus porous copper was produced within a hydrogen ratio range of 25-50%, a transference velocity range of 30-90 mm/min, and a total pressure range of 0.2-0.4 MPa. As a result, the porosity ranged from 36% to 55% and the pore size varied from 300 to 1500 µm, demonstrating a wide range of porosity and pore size. Through process optimization, it is possible to control the de-sired porosity and pore size.
Line 346-453
Particularly important and good in my opinion is chapter 4.2 Pore diameter. I do not have any comments on this area, but I pay attention to commend the authors.
I congratulate the authors on the presentation of this issue.
Response 2
I sincerely appreciate the reviewer's compliments.
Chapter 2 Materials and Methods
Line 85
Because of the solubility of hydrogen and oxygen in copper and the occurring relationship in this regard, the authors could have provided the oxygen content of the copper used for the process. The oxygen present in the copper could have meanings on the obtained effects of the analyzed process.
Response 3
I am aware that hydrogen gas is partially used in copper deoxidation processes, where oxygen reacts with hydrogen to form H2O, leading to deoxidation. However, this paper did not consider the deoxidation reaction and its potential influence on pore formation. It has been confirmed that the deoxidation reaction plays a different role in pore formation. Nevertheless, this study did not address the impact of the deoxidation reaction on the pore structure. Currently, I am working on a new paper that explores the changes in pore structure resulting from this confirmed deoxidation reaction.
Chapter 3 Results
This chapter is, in my opinion, complete. The pictures are of good quality, the graphs show the changes in the specified parameters. Only the colors on the graphs could be standardized or the use of photos of copper in real colors. However, these are details that are not important. One can confidently disregard these comments.
Response 4
It was observed that the color of the copper varied slightly depending on the polished surface. To achieve a uniform appearance, the color was adjusted to gray for consistency.
Line 407
“It has also been reported that increasing the solidification rate reduces the diffusion 407 time and inhibits pore growth [15].”
This is self-explanatory.
Response 5
It has been removed.
Chapter 5 Conclusions
The summary is good, but the chapter could be supplemented with specific parameters to be used to achieve certain effects.
Response 6
I have revised the conclusion overall.
Process Parameter Optimization: Optimization of process parameters is essential to control the pore structure of lotus porous copper. To achieve the smallest pore size, a hydrogen ratio of 25%, a feed rate of 90 mm/min and a total pressure of 0.4 MPa resulted in a porosity of about 36% and a pore size of 300 µm. By systematically adjusting the process parameters within the ranges of a hydrogen ratio of 25-50%, a transference velocity of 30-90 mm/min, and a total pressure of 0.2-0.4 MPa, it is possible to precisely control the porosity (36-55%) and pore size (300-1500 µm) of lotus porous copper. This makes it possible to produce materials with properties tailored to specific requirements.

Reviewer 3 Report
Comments and Suggestions for Authors
The authors have addressed a significant topic to help industrial fabrication of lotus porous copper parts by continuous casting. However, some aspects need attention from the authors for further resubmission, as the current state of the manuscript needs mayor revision
1. Abstract is confusing. The influence of the experimental parameters over the porosity and pore diameter are not clearly explained. Lines 13-18 mention such influence but it is dificult to follow the main idea.
2. I strongly recommend to change the term transference velocity for cooling rate (ºC/min). This will represent more clearly the solidification system you used.
3. Experimental condition listed in table 1 are single operations or, it was a systematic evaluation of each parameter? Meaning 3x3x3:27 experiments. Please clarify it.
4. Table 1 should be display all the experiment conducted.
5. Line 110-111: Section were cut from the original bar at with position. Please identify it.
6. Section 3.2: Throughout this section, the experiments were conducted changing 2 parameters simultaneously during the experiment or, it just the analysis of discrete experiment. Please clarify.
7. Equation 3 and 4 are basically the same. Please remove one.
8. Line 342: A comparative graph of the theoretical and measured values of the porosity worth to be included in the manuscript.
9. Line 377: If pores are mainly elongated, how does the author estimated the average pore volume? Please elaborate.
10. Figure 12b: It is this number of detected pores calculated or is the experimental value. Please clarify it.00
11. Why do the authors not include a figure to compare experimental and theoretical values of detected pores?
12. Suggestion: If mainly two variables were measured: % of porosity and pore diameter. Does the author explore the use of XYZ graph? As Rh (X), Vc (Y), %porosity or pore diameter (Z), and Pt as different surface.
Author Response
Dear Reviewer
The first feedback taught me a lot. I really appreciate it. We considered them carefully and tried to reflect them in the revised manuscript accordingly.
Following are the summaries of our revision. I don't think all of the reviewer's opinions have been reflected. If it has not been reflected or if additional corrections are required, please kindly let us know. Looking forward to hearing from you soon.
Best regards,
Byungsue shin
E-mail: 22202189@inha.edu
Response to Reviewer 3 Comments
The authors have addressed a significant topic to help industrial fabrication of lotus porous copper parts by continuous casting. However, some aspects need attention from the authors for further resubmission, as the current state of the manuscript needs mayor revision
- Abstract is confusing. The influence of the experimental parameters over the porosity and pore diameter are not clearly explained. Lines 13-18 mention such influence but it is dificult to follow the main idea.
Response 1
I have made the revisions.
Lotus porous copper was produced within a hydrogen ratio range of 25-50%, a transference velocity range of 30-90 mm/min, and a total pressure range of 0.2-0.4 MPa. As a result, the porosity ranged from 36% to 55% and the pore size varied from 300 to 1500 µm, demonstrating a wide range of porosity and pore size. Through process optimization, it is possible to control the desired porosity and pore size.
- I strongly recommend to change the term transference velocity for cooling rate (ºC/min). This will represent more clearly the solidification system you used.
Response 2
I agree that expressing it in terms of solidification rate is the clearest and most accurate approach. However, since it is difficult to directly measure the solidification rate in the continuous casting equipment, we used the transference velocity as an indirect indicator. Please note that a faster transference velocity implies a faster solidification rate. Thank you for your understanding.
- Experimental condition listed in table 1 are single operations or, it was a systematic evaluation of each parameter? Meaning 3x3x3:27 experiments. Please clarify it.
Response 3
This experiment was indeed conducted 27 times, following a 3x3x3 experimental design. Accordingly, I have updated Table 1 to reflect this adjustment.
- Table 1 should be display all the experiment conducted.
Response 4
It has been revised.
- Line 110-111: Section were cut from the original bar at with position. Please identify it.
Response 5
I apologize for the confusion. I didn't realize that the bar or position was cut off in lines 110 and 111. I will review and correct those sections accordingly.
- Section 3.2: Throughout this section, the experiments were conducted changing 2 parameters simultaneously during the experiment or, it just the analysis of discrete experiment. Please clarify.
Response 6
It has been revised as follows.
The effects of two or three process parameters on pore structure were analyzed by varying these parameters in different experimental sets. In each experiment, either one of the parameters (hydrogen fraction, transference velocity, or total pressure) was fixed while the other parameter was varied, or all three parameters were varied simultaneously.
- Equation 3 and 4 are basically the same. Please remove one.
Response 7
It has been revised.
- Line 342: A comparative graph of the theoretical and measured values of the porosity worth to be included in the manuscript.
Response 8
The theoretical porosity were calculated using Eq. 10, and the comparison between the theoretical and measured porosity is displayed in Figure 2. The relevant details have been included in the figure caption.
- Line 377: If pores are mainly elongated, how does the author estimated the average pore volume? Please elaborate.
Response 9
I have additionally included the following content.
The porosity and pore diameter of samples manufactured by the continuous casting show more consistent trends compared to other manufacturing methods. As a result, the average pore volume can be estimated from the porosity and the number of detected pores. Although the elongated shape of the pores may cause slight variations in the number of pores depending on the cross-sectional position of the specimen, the variations in pore size and number of pores are minimal.
- Figure 12b: It is this number of detected pores calculated or is the experimental value. Please clarify it.00
Response 10
It has been revised.
- Why do the authors not include a figure to compare experimental and theoretical values of detected pores?
Response 11
The number of detected pores is influenced by various factors, such as pore nucleation and the thermal conductivity diffusion of the material, making it challenging to accurately predict the actual number of pores. Therefore, it is difficult to directly compare experimental and theoretical values. In this study, the goal was to explain the trends in pore nucleation and growth based on the number of detected pores.
- Suggestion: If mainly two variables were measured: % of porosity and pore diameter. Does the author explore the use of XYZ graph? As Rh (X), Vc (Y), %porosity or pore diameter (Z), and Pt as different surface.
Response 12
Thank you for the valuable suggestion. However, presenting the data using an XYZ 3D graph makes it difficult to clearly observe the trends under various process conditions. For this reason, we have opted to use an XY 2D graph for better clarity

Round 2
Reviewer 3 Report
Comments and Suggestions for Authors
All the points were resolved adequately.
The paper can now be accepted for publication.